# An Evaluation and Promotion Strategy of Green Land Use Benefits in China: A Case Study of the Beijing–Tianjin–Hebei Region

**Wenying Peng [1,\*], Yue Sun [1], Yingchen Li [1,2] and Xiaojuan Yuchi [1]**

1    School of Urban Economics and Public Administration, Capital University of Economics and Business, Beijing 100070, China; sjmqjsy@cueb.edu.cn (Y.S.); liyingchen0325@163.com (Y.L.); 12020010002@cueb.edu.cn (X.Y.)

2    Beijing Municipal Commission of Planning and Natural Resources, Beijing 101160, China

\*    Correspondence: cjpengwy@cueb.edu.cn

**Abstract:** Green development is the inevitable choice for global sustainable development, and China has chosen green development as its national strategy. Land use changes will affect a soil's organic matter by changing the land's productivity, soil quality and fertility. It is of great significance for ensuring soil fertility, improving the environment and promoting the carbon cycle that the concept of green development is implemented in the process of land use activity. Establishing an indicator system and evaluation method for a green land use benefit evaluation suitable for green development is helpful for strengthening the responsibility and consciousness of such land use, and to provide theoretical guidance and decision-making references for promoting such developments and evaluations. In this study, based on a connotation analysis of green land use, the entropy weight method and BP (Back Propagation) neural network model method were used to construct an evaluation index system for green land use benefits, including four criterion layers and eighteen evaluation indexes, and the entropy-BP neural network evaluation method was proposed to reveal the problems in green land use benefits in the Beijing–Tianjin–Hebei region. The results showed that the green land use benefit level in the region was low, while the spatial pattern was high in the north, low in the middle and high in the south. Langfang, Beijing and Handan were the lowest centers of green land ecological benefit, while Beijing and Tianjin were the lowest centers of green land economic benefit. The green governance benefit and green space benefit were in a relative spatial equilibrium. The cultivated land area, forestry products, sewage centralized treatment degree and built-up area ratio were the most important influences on the green ecological benefit, green economic benefit, green governance benefit and green space benefit, respectively. The entropy-BP neural network evaluation system and method have certain applications in the design of relevant assessment reward-and-punishment systems. Accelerating the optimization of the Beijing–Tianjin–Hebei territorial space's development and utilization pattern, and constructing a green benefit sharing mechanism of land use, are important strategies to improve the benefits of green land use.

**Keywords:** green development; green land use benefits; entropy-BP neural network evaluation; regional synergy

## 1. Introduction

With the rapid industrialization and urbanization of the world, land use/land cover change is increasingly intensifying and changing with new developments [1]. Land itself is a dynamic system, a form of material resources, and the resource that brings the greatest benefits to humans in the ecosystem [2]. In 1995, the "International Geosphere-Biosphere Programme" and the "International Human Dimension Programme on Global Environmental Change" were put forward, and the relationship between the human driving force's feedback with the environment began to be studied, deepening the relationship between land, population, and development, improving the utilization efficiency of land resources,

and guiding the rational use of land resources. Research on land use and land cover dynamics has become a key focus across the world [3,4]. Meanwhile, different land use strategies bring different land benefits to different regions, mainly including economic, social, and ecological benefits [5], and for a long time, research on land use benefits have received much attention. Some researchers have discussed land use efficiency and its regional patterns from the perspective of a comprehensive efficiency. For example, the temporal and spatial characteristics of land use benefits in China during 2004–2014 were quantitatively analyzed from the perspectives of economy, society, and ecology [6]. The benefits of land use have been studied from the perspective of land use type. For example, a linear programming model has been used to evaluate the efficiency of agricultural land use [7]. Moreover, using slope data calculated from the "Shuttle Radar Topography Mission Digital Elevation Model" data set and multi-period land cover data from China's "Multi-Period Land Use Land Cover Remote Sensing Monitoring" dataset, we revealed the spatiotemporal pattern evolution of vertical urban expansion in China from 1990 to 2015 [8], while urban land use efficiency and its change were quantitatively analyzed from the perspectives of an ecological environment and social environment change [9]. From different administration levels, or specific management areas used to study the land use benefits, such as China's Yangtze River Delta, the Pearl River Delta and the Beijing–Tianjin–Hebei urban agglomeration, the urban comprehensive benefits of land use and its internal coupling coordination were studied [10]. This was based on urban and rural integration development and urbanization's healthy development to build an urban land use efficiency evaluation system [11], which was used to evaluate the spatial pattern characteristics of the economic, social, ecological and comprehensive benefits of land use at the provincial level [12]. With this study of land use/land cover, the land use benefit evaluation has become mature in both its theory and method.

In recent years, how to cope with global change, how to promote green development to improve the quality of ecosystem services, how to achieve a carbon peak, carbon neutrality, and other issues, have attracted increased attention. Continuing to deepen sustainable development, green and ecological benefits have become some of the main goals in the study of land use benefits. Land use is closely related to population growth, urbanization quality and environmental issues [13,14] and based on the connotation of land use benefit, some researchers quantified the green land use benefit of 284 cities at the provincial level and above in China based on green development goals, and pointed out that there are obvious spatial differences [15–17]. Studies from urban and rural land cover perspectives point out the differences in ecosystem functions between urban and rural areas [18], while the level of green land use has been evaluated from the perspective of the coupling coordination between land use and economic growth [19]. From the perspective of carbon storage, land use patterns and ecological benefits are also studied, and it is believed that land use patterns affect the carbon storage capacity of urban land, while the ecological benefits of different land use patterns are significantly different [20]. Based on the management of ecological protection red lines, the ecological benefit of land use was simulated and evaluated, with the ecological benefit of ecological protection red line areas being significantly improved [21]. The ecological green equivalent principle and method were used to evaluate the ecological service functions of different land cover types, and it was demonstrated that the ecological benefits of land use differ significantly under different land management modes [22]. Moreover, other scholars have used multi-scenario simulations of urban land use to reveal the contradiction of urban land use, and found that urban development is more sustainable under ecological and farming constraints. Based on scholars' discussions on a land use benefit evaluation, in China, this mainly focuses on the land use ecological benefit or the land use social benefit, and there is not a complete system for the evaluation of the green land use benefit [23]. In recent years, with the deepening of sustainable development and green development research, increasing attention has been paid to the study of green land use benefits. Based on the existing empirical research, this paper discusses the evaluation and promotion strategies of a green land use benefit.

Over the past 20 years, China has continuously strengthened the restoration of its ecosystems and has achieved remarkable ecological results in forest, grass, and wetland areas, making contributions to global ecological security and sustainable development [24]. Many previous studies evaluated land use changes and their associated benefits and more mature theoretical methods have been developed in terms of index system construction and quantitative analysis, such as the analytic hierarchy process (AHP), the input–output analysis (DEA) and land system dynamics. With the development of a complex network system, some scholars have considered land use/land cover as a human–environment system, which is an important factor in terms of constructing a conceptual model of a land system [25], the introduction of the neural network model for the regional development level [26], sustainable development [27] and land suitability evaluations [28], cultivated land quality evaluations [29], etc. The research methods present a trend from a single method to a comprehensive application of multiple methods [30]. Green development is a model that emphasizes the interaction between the economy and the environment [31], including three first-level indicators of the green development degree of economic growth, resource and environmental carrying capacity, and government policy support, and nine second-level indicators of green growth efficiency, resource abundance and ecological protection, and green investment. Additionally, there are 62 third-level indicators of a green development index system, including the regional total output value per capita, forest coverage rate, and urban per capita green area. [32]. An "object-subject-process" framework system for global green development can be constructed under the framework of the Sustainable Development Goals (SDGs) [33]. The land use type and efficiency reflect the level of intensive resource utilization and ecological and environmental protection, leading to differences in the ecological and environmental effects which will affect the construction of regional ecological civilization and the level of green and high-quality development [34,35]. Eventually, this will significantly affect the surface $CO_2$ concentrations [36]. Nowadays, land use change and its ecological, environmental problems should be measured from the perspective of ecological civilization construction and green and high-quality development [37]. The green development of land use should strengthen our understanding of the resources, capital, and assets of land systems, and help us to better grasp their ecological services, ecological assets, multi-functional landscapes, and ecological security [38]. Some scholars have also used the Theil Index and spatial Markov chain to measure the spatial patterns and to analyze the evolution of green development efficiency in Chinese cities from 2005 to 2015. Based on the above methods of assessing the land use change, this paper uses the entropy weight method and BP neural network method for research and constructs the entropy weight–BP neural network model of green land use benefits. The initial result calculated by the entropy weight method was used as the input of the BP neural network, where the connection weight between each layer was learned by the neural network and the weight was calculated, making the results of the land resource green benefits more scientific and objective [39]. Existing studies have mostly evaluated and analyzed land use change and its benefits, and the ecological and environmental effects, etc.; however, it is necessary to further promote the comprehensive evaluation of green development and land use benefits to provide a theoretical basis for rationally optimizing land use and promoting green development in land management.

Green land use introduces the concept of green development into the process of land use, which is also the goal of land use [16], whereas green land use resources refer to the minimum input of land resources to obtain the maximum social, economic and ecological benefits, which have an important impact on global sustainable use and high-quality development. The green utilization of land resources requires an objective and comprehensive understanding of the status quo, the quality and problems of regional land resources, a systematic analysis of the supply and demand for land resources, and a rational organization of land productivity and distribution, which ensures the coordinated operation and sustainable development of a land system and that constantly improves the ecological and economic functions of that land. Thus, we can then obtain the economic,

ecological, and social benefits of a land use type [40]. A green benefit is a comprehensive benefit involving the economy, environment, society, and ecology [41]. Capital, labor, and energy are often taken as the input factors, green economy as the expected output, and environmental pollution as the unexpected output to measure the green development benefits [42]. Land resources have natural, economic, ecological, and social attributes and they are not only the means of production but also the objects of labor. Land use is a comprehensive system that can better reflect green benefits and based on the existing research and following the concept of green development, the green land use benefit can be composed of four subsystems: the green ecological benefit, green economic benefit, green governance benefit and green space benefit. From the perspective of an ecosystem service difference caused by differences in the land use type, a green ecological benefit emphasizes the overall benefit level of an ecosystem service. From the perspective of the most basic production function of the land, agricultural primary products produced by that land bring economic benefits to the region; therefore, green economic benefits reflect green products and their benefits. In the process of land development and utilization, affected by human factors such as policies and capital investment, green governance benefits emphasize the benefits brought by a government's governance of the ecological environment. From the perspective of the adjustment of urban green space brought about by territorial space optimization, green space benefits emphasize the level of green space service benefits provided by the land for human recreation; therefore, this article is based on green development and the connotations of green land use. The approach is based on the integrated use of the entropy method and BP (Back Propagation) neural network model. These techniques can quantitively and robustly reveal the Beijing–Tianjin–Hebei region's green land use efficiency and its influencing factors in order to explore the optimum path of the region's land use and land-use- and ecological-environment-coordinated development of policies and measures. It lays a theoretical foundation for the green and high-quality development of land management in the Beijing–Tianjin–Hebei region.

## 2. Materials and Methods

### 2.1. Evaluating the Green Benefits of Land Use

To objectively measure the level of green land use benefits, this paper constructed an entropy-BP neural network evaluation model of green land use benefit, as shown in Figure 1. Firstly, based on the connotations of the land use system and green benefit, the evaluation index system was established according to four subsystems: the green ecological benefit, green economic benefit, green governance benefit and green space benefit. Secondly, the entropy weight method was used to standardize the index and determine the weight, and the initial score of the green land use benefit in the region was obtained. Thirdly, the weights and scores of various indexes obtained by the entropy weight method with certain objectivity were taken as the input and output of the BP (Back Propagation) neural network. The weights between neurons were used to obtain the weight of the green benefit index of land use, and the index was calculated. Finally, the index was used to analyze the spatial patterns and land use optimization and to put forward the strategy to improve the green land use benefit. The analytical thinking and model construction are shown in Figure 1.

### 2.2. Research Methods and Index System

#### 2.2.1. Entropy Method and BP Neural Network Method

The entropy weight method is an objective weight assignment method based on entropy values. It can directly and objectively reflect the average situation of index energy distribution by calculating the entropy value through an index variation, obtaining the corresponding weight through correction, and it can accurately evaluate the importance of a related index. In a comprehensive evaluation, the entropy value can be used to judge the importance of an index. When the data are more dispersed, the entropy value is smaller; therefore, the data contain more information and the weight is larger. The greater the dispersion degree of the index, the greater the influence of the index on the

comprehensive evaluation, that is, the greater the weight; however, the entropy weight method is highly dependent on sample data, for example, there is no significant unity in the index weight, and the weight obtained is the current useful information of the index, which may cause errors. When the accuracy of the analysis results is high, it is necessary to find a method with less errors. The indicators selected in this paper included forward indicators and reverse indicators. Forward indicators mean that the higher a value is, the better the evaluation is, while reverse indicators are the opposite; therefore, it was necessary to reverse the backward indicators to ensure the indicators had the same trend, and then normalize the data. The BP (Back Propagation) neural network was a multilayer feedforward neural network. The network layers were related to each other by a connection weight coefficient, and the neurons in the same layer were independent from each other. The optimal value was obtained according to the adaptive gradient descent method, and the square of the network error was the target. The extension structure of the BP neural network model included an input layer, hidden layer, and output layer, as shown in Figure 1. Using the combination of the entropy weight method and the BP neural network model to calculate the index weight, the evaluation index improved the accuracy and precision of the results and better reflected the objectivity of the evaluation results. The specific steps are as follows:

(1) Before using the entropy weight method, due to the different dimensions and units of each index, it is impossible to directly compare and calculate them; therefore, the data are standardized. The matrix X represents m evaluation samples and n indicators:

$$X = \left(x_{ij}\right)_{m*n} (i = 1, 2, \cdots, m, j = 1, 2, \cdots, n)$$

When the indicator is a forward indictor, its standardized formula is:

$$x'_{ij} = \frac{x_{ij} - x_j^{min}}{x_j^{max} - x_j^{min}} \tag{1}$$

When the indicator is a reverse indictor, its standardized formula is:

$$x'_{ij} = \frac{x_j^{max} - x_{ij}}{x_j^{max} - x_j^{min}} \tag{2}$$

(2) The entropy weight method is used to calculate the initial index weight of the green land use benefit, and the following initial information matrix is obtained:

$$e_j = -\frac{\sum_{i=1}^{m} y_{ij} \ln y_{ij}}{\ln m} \tag{3}$$

$$\omega_j = \frac{1 - e_j}{n - \sum_{i=1}^{n} e_j} (i = 1, 2, \cdots, m, j = 1, 2, \cdots, n) \tag{4}$$

$$W = \left(\omega_j\right)_{1*n} \tag{5}$$

In the Formulas (3)–(5), $e_j$ represents the entropy value of the evaluation index, n is the number of index items, $\omega_j$ represents the entropy weight of the index j, and W is the initial total weight of the system.

(3) Using the BP neural network training process model, the learning rate is set as 0.1, the activation function uses the unipolar Sigmoid function, and the training function uses the Levenberg–Marquardt function (trainlm function), which are all available in the toolbox of the MATLAB system. The neural network learning algorithm is established to determine the decision weight of the input factors to output factors. The formula is as follows:

$$\omega_i = \frac{\sum_{l=1}^{k} \left|v_{jl}\right|}{\sum_{i=1}^{m} \sum_{l=1}^{k} \left|v_{il}\right|} \quad j = 1, 2, \ldots, m \tag{6}$$

where $v_{il}$ and $v_{jl}$ represent weights from the neuron i to layer l and neuron j to l, respectively, namely, the elements in matrix V and Z. $\omega_i$ represents the i index weight in the evaluation system of the green land use benefit obtained by the BP neural network.

(4) To quantitatively evaluate the green benefits of land use, the entropy-BP neural network model of green benefits of land use is constructed by combining the entropy-BP neural network model with the BP neural network model. The calculation formula is as follows:

The green benefit indices of the four criteria layers are as follows:

$$g_{jz} = \sum_{i=1}^{m} \omega_i * x'_{ij}, z = 1, 2, 3, 4 \tag{7}$$

The total land use green benefit index of an output layer is:

$$G_j = \sum_{i=1}^{n} \omega'_i * g_j \tag{8}$$

where $x'_{ij}$ represents the normalized value of index i, $\omega_i$ represents the weight of index i, and $\omega'_i$ represents the weight of the criterion layer i. $g_{jz}$ represents the green benefit index of the criterion layer, and $G_j$ represents the green benefit index of the land use.

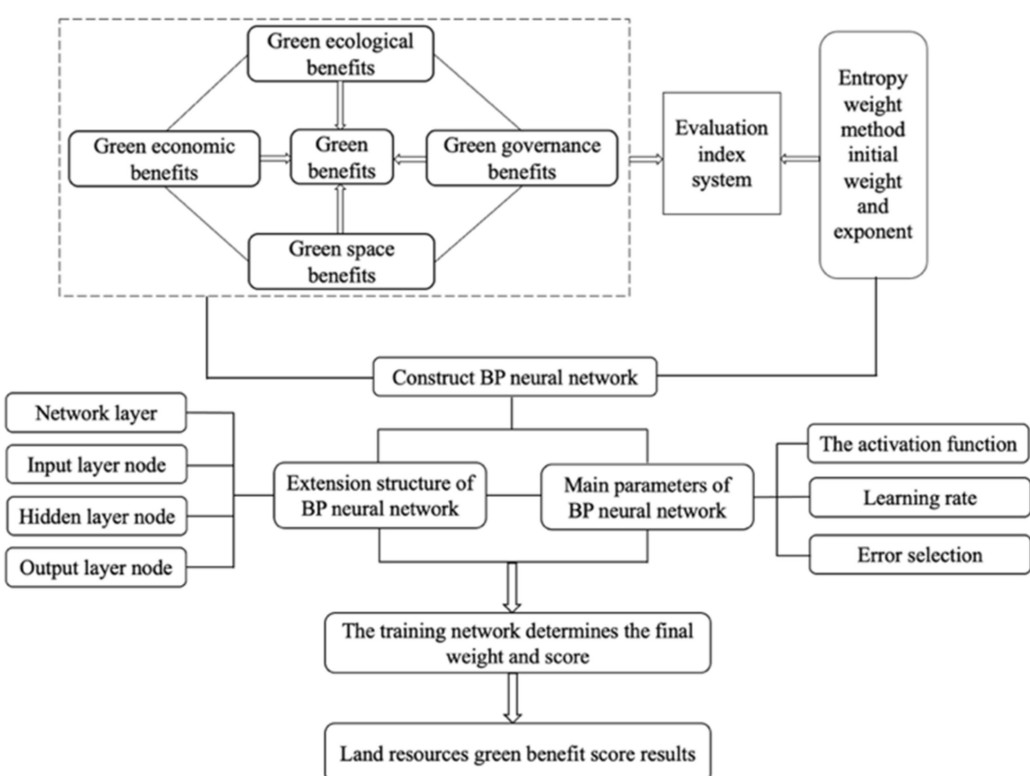

**Figure 1.** Theoretical framework and evaluation model of green land use benefit.

2.2.2. Evaluation Index System of Green Land Use Benefit

Land use is a highly complex system in which human activities and the natural ecology interact and influence each other. The green benefit reflects the comprehensive goal of a coordinated development of natural ecology and economic society. In this paper, the land resources outputs of the ecology, economic governance and environment, of green land use benefits—considering that a land resource has natural, economic, ecological and social attributes—are the human means of production and labor objects, namely, a comprehensive system for producing green benefits. The evaluation index system of the green land use benefit was determined as a three-layer structure of a target, criterion, and index. The top layer is the target layer, that is, the goal of the index system is to evaluate the pattern of a regional land's green benefit. The middle layer is the criterion layer, which makes the evaluation of regional land green development more specific. The last layer is the index layer, which is the expansion of the middle layer. The factors affecting the regional land green benefit outputs were then analyzed in a more comprehensive and specific way.

The target layer of the evaluation index system was the green land use benefit, and the four criterion layers were the land's ecological benefit, land's economic benefit, land's governance benefit and land's space benefit, respectively.

After determining the hierarchy of the land use green benefit evaluation index system, the specific index content of the index layer was selected. After determining the initial direction in accordance with the principle of constructing the indicators mentioned above, the indicator system was determined by referring to the relevant literature. Some scholars took 14 cities in China along with the social benefit, economic benefit and ecological benefit of land use to evaluate the comprehensive benefits and ecological benefits including, the greenbelt area per capita, green coverage, etc. The economic indicators mainly included economic development indicators, such as the GDP, per capita GDP, and others. The social indicators included those focusing on urban residents' quality of life, such as the urban per capita road area and urban per capita water consumption [43]. When evaluating the land's ecological benefits, Li took into account the control rate of land desertification pollution, the control rate of soil erosion, an increased rate in per capita green space, etc. [44]. The social benefit index included the added value of agricultural labor products and the increased rate in per capita cultivated land area. The economic benefit index included an increased rate in cultivated land area and increased rate in yield per unit area. Shi evaluated the land use comprehensive benefits from the perspectives of a scale benefit, structural benefit and intensive benefit, among which the scale benefit involved the per capita construction land scale and construction land scale per unit GDP [45]. The structural benefits involved the proportion of cultivated land area and woodland area, and the intensive benefits involved the investment of secondary and tertiary industries. Liang evaluated land use benefits from the two perspectives of the social and economic benefits and the ecological and environmental benefits, including indicators such as the population density in built-up areas, the per capita living area of urban residents, the green coverage rate in built-up areas, the forest coverage rate, the industrial wastewater treatment rate, and the proportion of environmental input in total investment [46]. Based on previous studies, this paper selected representative, comparable and objective indicators to complete the construction of the index system of study.

In the criterion layer of the green ecological benefit of land, there were five specific indicators, namely, the per capita forestland area, per capita grassland area, per capita wetland (including water area) area, per capita cultivated land area and per capita water resource amount. In the criterion layer of the land's green economic benefit, there were four specific indicators, namely, the per capita output value of agricultural products, the per capita output value of forestry products, per capita output value of animal husbandry products and per capita output value of fishery products. In the criterion layer of the land's green governance benefit, there were four specific indicators, which were the effective irrigation area, the soil and water loss control area, centralized sewage treatment rate and per capita urban environmental infrastructure construction funds. In the criterion layer of the land's green space benefit, there were five specific indicators, which were the proportion of a nature reserve area, forest coverage rate, park green land rate, land utilization rate and built-up area ratio. We collected the evaluation index and standardized it using the entropy weight method formula (4) (5) to calculate the initial weights of each index and criterion layer. Then, we input the weight into the training process of the BP neural network model (6), and using a neural network model correction we paired each index system. The contribution of the BP network to modifying the initial weight after the final weight is shown in Table 1.

**Table 1.** Green land use benefit index system and BP network index weight in Beijing–Tianjin–Hebei region.

| Target Layer | Criterion Layer | Weight | Index Layer | Unit of Index | Direction | Weight |
|---|---|---|---|---|---|---|
| Land use green benefits | Green ecological benefit | 0.2675 | Forest area per capita | Ha/ten thousand people | + | 0.1985 |
| | | | Grassland area per capita | Ha/ten thousand people | + | 0.1844 |
| | | | Per capita area of wetland (including water area) | Ha/ten thousand people | + | 0.1685 |
| | | | Per capita cultivated area | Ha/ten thousand people | + | 0.2350 |
| | | | Water resources per capita | Cubic meters/person | + | 0.2135 |
| | Green economic benefit | 0.2355 | Per capita output value of agricultural products | CNY/person | + | 0.2259 |
| | | | Per capita output value of forestry products | CNY/person | + | 0.2728 |
| | | | Per capita output value of animal husbandry products | CNY/person | + | 0.2557 |
| | | | Per capita output value of fishery products | CNY/person | + | 0.2455 |
| | Green governance benefit | 0.2416 | Effective irrigation area | Thousands of hectares | + | 0.2393 |
| | | | Soil erosion control area | Thousands of hectares | + | 0.2284 |
| | | | Centralized sewage treatment rate | % | + | 0.3306 |
| | | | Per capita urban environmental infrastructure construction funds | CNY/person | + | 0.2017 |
| | Green space benefits | 0.2555 | Proportion of nature reserve area | % | + | 0.1844 |
| | | | Forest coverage | % | + | 0.2102 |
| | | | Green rate of park | % | + | 0.1485 |
| | | | Utilization rate of land development | % | − | 0.2240 |
| | | | Built-up area ratio | % | − | 0.2330 |

Note: "+" and "−" represent the positive and negative indicators, respectively. "CNY": is the unit of measurement for China's legal currency.

### 2.3. Study Area and Data Source

2.3.1. Overview of the Study Area

The Beijing–Tianjin–Hebei region is located at 36°01′~42°37′ N, 113°04′~119°53′ E, bordering Bohai Bay in the east, Shanxi province in the west, Inner Mongolia in the north and Shandong province and Henan province in the south. The total area is 213,600 square kilometers, accounting for 2.3% of China's territory. The specific geographical location of the Beijing–Tianjin–Hebei region is shown in Figure 2. The terrain is mainly mountainous and plain, with Taihang Mountain in the west, Yanshan and Bashang Plateau in the north, and Haihe plain in the remainder. Specifically, Beijing is dominated by mountainous areas, accounting for about 62% of the area. Tianjin is dominated by plains, which account for more than 90%. Hebei province consists of the Bashang Plateau, Yanshan, Taihang Mountain and Hebei plain, accounting for 8.5%, 48.1% and 43.4% of the total area of the province, respectively.

The Beijing–Tianjin–Hebei region is the largest region in northern China with the strongest comprehensive strength. It is a key area for green development and an important frontier for coordinated regional economic development in China. At present, the region is dominated by high-pollution and high-energy industries, the ecosystem is very fragile, and the contradiction between humans and the land is prominent. The region was taken as the research area to evaluate the level of green land benefit and explore the development pattern of green land resource benefits in the region. It would be helpful to solve the ecological, environmental problems in the Beijing–Tianjin–Hebei region, to realize the sustainable utilization of land resources and to improve the utilization efficiency of its land resources.

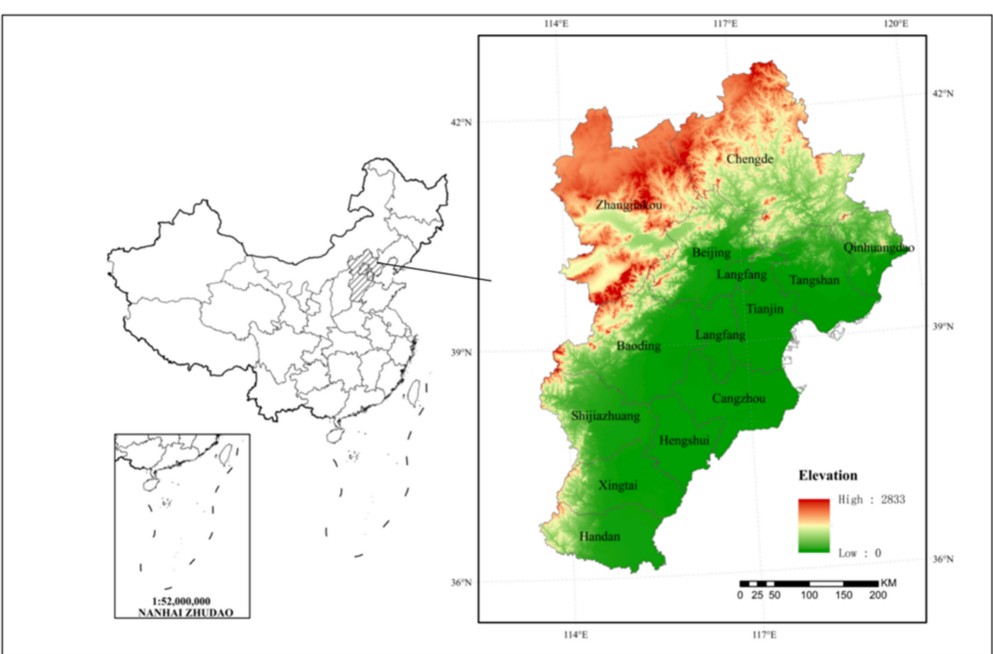

**Figure 2.** Location of the Beijing–Tianjin–Hebei region.

According to the Ministry of Housing and Urban–Rural Development of the "Beijing-Tianjin-Hebei urban agglomeration coordinated development program 2008–2020", it is the national political and cultural center of the Beijing–Tianjin–Hebei region, including Beijing, Tianjin, Baoding, Hebei province, Langfang, Shijiazhuang, Tangshan, Handan, Qinhuangdao, Zhangjiakou, Chengde, Cangzhou, Xingtai, and Hengshui. By 2021, the region accounted for 7.81 percent of China's population, 69.3 percent of its urbanization and 8.4 percent of the GDP. According to the *Statistical Yearbook of China's Urban and Rural Construction* for 2020 and the third *Land Survey* and *Data Bulletin of Beijing*, the region accounted for 10.9 percent of the country's construction land. The per capita arable land area was 0.06 ha, lower than the national average, and most of this was in the Huairou, Yanqing and Miyun districts of Beijing and Chengde, and Zhangjiakou and Baoding of the Hebei province. The water resources per capita were 168 cubic meters, much lower than the national average, and the water area accounted for 2.6%, mainly in Tianjin and the Tangshan and Cangzhou cities of the Hebei province.

### 2.3.2. Data Sources

The data required for this study came mainly from the National Bureau of Statistics of China, the Beijing Municipal Bureau of Statistics, the Tianjin Municipal Bureau of Statistics of Hebei Province and other relevant websites. This included the *China Urban Statistical Yearbook 2018*, *China Environmental Statistical Yearbook 2018*, *China Rural Statistical Yearbook 2018*, *Beijing Statistical Yearbook 2018*, *Tianjin Statistical Yearbook 2018*, *Tianjin Water Resources Bulletin 2018*, *Hebei Economic Yearbook 2018*, *Hebei Rural Statistical Yearbook 2018*, the *Baoding, Cangzhou, Hengshui and 11 other prefectural cities in Hebei provinces' Statistical Yearbook*, and national economic and social development statistical bulletins. All the data were standardized first, and the standardized data was between 0 and 1, which was convenient for the subsequent calculations.

## 3. Results and Discussion

This paper, based on the entropy-BP neural network model, obtained the Beijing–Tianjin–Hebei green land use efficiency index, green efficiency levels, and the internal structure and regional structure, revealing the green benefit influence factors, based on the concept of "ecological priority and green development", promoting the development and

utilization of national spatial structure optimization and a regional coordinated development, thus proposing a promotion strategy of the green benefits.

### 3.1. Contribution of Green Benefit Index of Land Use

The weight of each index reflected its contribution to the green efficiency index, and then reflected the impact of each index on the regional pattern of green efficiency. The analysis of the index weight is shown in Table 1. This demonstrates that the green economic benefits, green ecological benefits, green governance benefits, and green space benefits had a relatively balanced contribution degree. The green ecological benefit and green space benefit contributed the most, accounting for 27% and 26%, respectively, while the green economic benefit contributed the least, accounting for 23%. The green benefit was the comprehensive benefit of the ecology, economy, society, and environment. For a region, the quantity of forest land, grassland, wetland, cultivated land and water resources is an important carrier of its ecosystem services. The per capita occupation degree largely indicates the regional ecological environment quality based on people. The proportion of natural reserve areas, forest coverage rate, and park green land rate, directly reflect the quality of green open spaces, while the utilization ratio of land development and built-up areas reflects the crowding of resources and the environment by urbanization and an economical and intensive use of land resources. Ecosystem services, ecological environment quality and resource conservation, and intensification are direct manifestations of the concept of "ecological priority and green development". Green development no longer only represents economic development, and ecological products are no longer only agricultural products. The contribution of the green benefit of the four subsystems of land use calculated by the entropy–BP neural network was in line with objective reality and target requirements.

In terms of the contribution of each index of the four subsystems, the index contribution of the green ecological benefit was in the order of per capita cultivated land area > per capita water resources > per capita woodland area > per capita grassland area > per capita wetland (including water area) area, accounting for 24%, 21%, 20%, 18% and 17%, respectively. In terms of the green economic benefits, the index contribution was forest product output value > per capita animal husbandry product output value > per capita fishery product output value > per capita agricultural product output value, accounting for 27%, 26%, 24% and 23%, respectively. In terms of the green governance benefits, the index contribution was the centralized sewage treatment rate > effective irrigation area > soil erosion control area > per capita urban environmental infrastructure construction funds, accounting for 33%, 24%, 23% and 20%, respectively. In terms of the green space benefit, the index contribution was the built-up area ratio > land utilization ratio > forest coverage ratio > nature reserve area ratio > park green space ratio, accounting for 23%, 22%, 21%, 19% and 15%, respectively. Overall, the cultivated land area, forestry products, sewage centralized treatment degree and built-up area ratios play the most important role in their subsystems, inspiring further analysis of the green benefit patterns in the Beijing–Tianjin–Hebei region. Cultivated land is the most basic resource and material condition for human survival, bearing important nurturing functions and ecosystem service functions at the same time; however, the per capita amount of the cultivated land resources in the Beijing–Tianjin–Hebei region is very limited, and the situation of cultivated land protection is severe. Consequently, it is of the utmost importance in terms of its green ecological benefits. As for irreplaceable ecological functions such as water conservation, soil and sand fixation, forests, and purifying the air, forestry is the main source of ecological construction, and economic forests and biological energy are used by the country to vigorously develop the green industry. For the mountainous areas of Beijing–Tianjin–Hebei, forestry products play an important role in the green economy. In the region, the water resources are in short supply, water pollution is high, and the effect of ecological water treatments is relatively slow. The larger the ratio of built-up areas in a region is, the more urban construction occupies the natural ecological space, and the more limited the quantity and quality of green space; therefore, the ratio of built-up areas can best reflect the efficiency level of green

space. The contribution of the index calculated by the entropy-BP neural network was relatively consistent with the objective situation, which also indicates it is an important factor for improving the level of the green land use benefit.

*3.2. Green Benefit Level of Land Use*

According to the evaluation index system and weight of the Beijing–Tianjin–Hebei land use green benefit, the green benefit index of land use was calculated by using the entropy-BP neural network model Formulas (7) and (8), as shown in Table 2. According to the results, the green benefit level of the land use in the region was low, with an average index of 0.376. From the internal composition of the green benefit, the green space benefit index was the highest, followed by the green governance benefit and green economic benefit, while the green ecological benefit index was the lowest. This shows that the green efficiency level of land use in the Beijing–Tianjin–Hebei region urgently needs to be improved. The efficiency level of the forest land, grassland, wetland, water resources and arable land per capita was low, and the output efficiency of green products per capita was not high. Since entering the 21st century, the Beijing–Tianjin–Hebei region has developed its ecological restoration and ecological protection, ecological agriculture, water-saving irrigated agriculture and regional coordination, it has promoted the prevention and control of atmospheric pollution and water pollution, and it has controlled soil pollution and prevented solid waste pollution. Its forest coverage has increased, and its green management and green space benefits are at relatively high levels.

**Table 2.** Green benefit index of land use in the Beijing–Tianjin–Hebei region.

| Study Area | Green Ecological Benefit | Green Economic Benefit | Green Governance Benefit | Green Space Benefit | Green Benefit |
|---|---|---|---|---|---|
| Beijing | 0.0791 | 0.1511 | 0.3150 | 0.5731 | 0.2792 |
| Tianjin | 0.2278 | 0.1700 | 0.1244 | 0.3981 | 0.2327 |
| Shijiazhuang | 0.1156 | 0.2694 | 0.6112 | 0.5378 | 0.3794 |
| Tangshan | 0.2570 | 0.5767 | 0.4497 | 0.5200 | 0.4460 |
| Qinhuangdao | 0.2894 | 0.7285 | 0.2665 | 0.6778 | 0.4865 |
| Handan | 0.0670 | 0.1957 | 0.4723 | 0.5293 | 0.3133 |
| Xingtai | 0.1374 | 0.1875 | 0.4543 | 0.4184 | 0.2976 |
| Baoding | 0.1490 | 0.2302 | 0.5111 | 0.6788 | 0.3910 |
| Zhangjiakou | 0.6522 | 0.5583 | 0.4418 | 0.4385 | 0.5247 |
| Chengde | 0.7978 | 0.5113 | 0.3719 | 0.6502 | 0.5897 |
| Cangzhou | 0.1749 | 0.2559 | 0.5114 | 0.4861 | 0.3548 |
| Langfang | 0.0997 | 0.2243 | 0.2106 | 0.4776 | 0.2524 |
| Hengshui | 0.1718 | 0.2963 | 0.5155 | 0.4137 | 0.3460 |

*3.3. Regional Pattern of Green Land Use Benefit*

As can be seen from Table 2, there were obvious regional differences in the green land use benefits in the Beijing–Tianjin–Hebei region, presenting a spatial pattern of high green land use benefits in the north, low green land use benefits in the middle, and high green land use benefits in the south. The green efficiency index of Hebei province was higher than that of Beijing, and that of Beijing was higher than that of Tianjin. Chengde and Zhangjiakou in the northern region had the highest green efficiency index of more than 0.5, mainly because of their rich forests, grasslands, wetlands and water resources, high forest coverage rate, low land development and utilization rate, and high agricultural output. The green benefit index of Tangshan and Qinhuangdao was 0.45 and 0.49, respectively, mainly due to high levels of agricultural products output, high forest coverage rates, low utilization rates of land development and built-up areas, which made their green economic benefits and green space benefits levels higher. Baoding, Shijiazhuang, Cangzhou and Hengshui had a green efficiency index between 0.32 and 0.39. These four cities are in the southern part of the region, and their green economic benefits are limited; however, urban green governance and an improvement in the spatial environment were relatively good. In

Handan, Xingtai and Beijing, the green benefit index was between 0.26 and 0.31, mainly because the total population is relatively large and the ecological benefit and agricultural output was very limited. The green benefit index of Langfang and Tianjin was the lowest, between 0.23 and 0.25, mainly because the total population is relatively large, the per capita level of forest, grass, wetland, and arable land was limited, the output of green agricultural products was not high, and the green governance and green space benefit levels were low.

Figure 3 shows the spatial pattern characteristics of the four subsystems. In the Beijing–Tianjin–Hebei region, where the land's green ecological benefits were higher in the north and the east, and lower in the middle and south. The green ecological benefit index of Chengde and Zhangjiakou was much higher than that of the other cities. The green ecological benefit index of Chengde was nearly 0.8; Qinhuangdao, Tangshan and Tianjin had higher green ecological benefits, with indexes ranging from 0.22 to 0.29. The index of Cangzhou, Hengshui, Baoding, Shijiazhuang and Xingtai ranged from 0.11 to 0.18; Langfang, Beijing and Handan had the lowest index, at below 0.1. This is consistent with the spatial pattern by which the northern region was rich in ecological resources, the eastern region was relatively rich in cultivated land resources and water resources, and the central and southern regions were relatively poor in ecological resources with more people and less land. At the top was Chengde's green ecological index, for example, making full use of the advantage of its forestry resources and its rich water resources to develop a "holistic approach to conserving mountains, rivers, forests, farmlands, lakes, and grasslands." as well as the in-depth implementation of the Beijing and Tianjin sandstorm source control, its pilot biomass waste management, the governance of small- and medium-sized rivers and so on; thus, forming the spatial pattern of an urban forest ecological system and obtaining the title of "the most beautiful ecological city in China". Conversely, the green ecological benefit index of Handan city was the lowest. Due to its long-term extensive development of industrialization and urbanization in the past, its vegetation coverage rate was low as was the per capita water resource level. Coupled with its small amount of wetland resources, the contrast between human activity and the land is prominent, which has affected the green ecological benefit level of its land utilization.

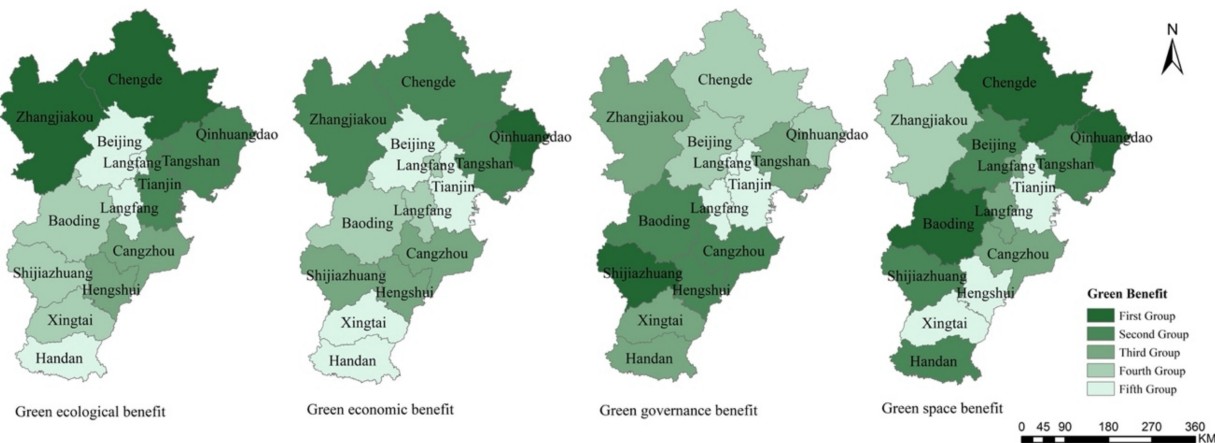

**Figure 3.** Spatial pattern of Beijing–Tianjin–Hebei land use green benefit subsystem.

The green economic benefit was higher in the north and south. Beijing and Tianjin had become low centers. Qinhuangdao had the highest green economic benefit index, which was as high as 0.73. The index of Zhangjiakou, Chengde and Tangshan ranged from 0.51 to 0.58, meanwhile, the index of Hengshui, Shijiazhuang, Cangzhou, Baoding and Langfang ranged from 0.21 to 0.3. Handan, Xingtai, Tianjin and Beijing had the lowest green economic efficiency index and this was consistent with the spatial pattern of the total population and output values of agricultural, forestry, animal husbandry, side-line, and fishery products. Qinhuangdao is in the Bohai Economic Circle; its coastal fishery was developed, it had low mountains, hilly areas and plains areas which are suitable for the

large-scale development of an aquaculture industry; therefore, its green economic benefits were prominent. Zhangjiakou and Chengde are rich in ecological resources, which have promoted the industrialization of forestry through the construction of the fruit-crop base and shrub processing industry, and its development of animal husbandry was good, which made great contributions to the local green economic benefits. Beijing and Tianjin had a high level of urbanization, a large population base, and a relatively low per capita output value of agriculture, forestry, animal husbandry, side-line, and fishery products, which depend on the supply of green products from foreign port bases.

The benefits of green governance were relatively low in the north and high in the south. Shijiazhuang had the highest index, as high as 0.61. The index values of Baoding, Hengshui and Cangzhou were around 0.51. In Handan, Tangshan, Xingtai and Zhangjiakou, the index was around 0.45. Chengde and Beijing's indexes were 0.37 and 0.32, respectively. The index of Qinhuangdao and Langfang was 0.27 and 0.21, respectively. Tianjin was the lowest, with an index of 0.12. Since the 18th National Congress of the Communist Party of China, unprecedented efforts have been made to promote the construction of ecological civilization, carrying out a series of ecological restoration and environmental remediation tasks, and generally paying attention to the benefits of green governance. Shijiazhuang's green governance efficiency was the highest, enhancing the comprehensive prevention and control of soil erosion, developing water-saving agriculture processes, optimizing agricultural water supply scheduling and other measures to improve on effective and low-irrigation farmland, and to encourage the combination of urban sewage concentrated treatment and disbursement methods to improve sewage treatment. Tianjin, which had the lowest green governance efficiency index, was mainly affected by soil erosion treatment areas and its urban sewage centralized treatment rate. Improving its sewage treatment rate will be an important measure in the future.

The regional distribution of the green space benefit was relatively balanced, with the indexes of Baoding, Qinhuangdao and Chengde exceeding 0.65 and that of Beijing, Shijiazhuang, Handan and Tangshan being between 0.5 and 0.57. Cangzhou, Langfang, Zhangjiakou, Xingtai, Hengshui and Tianjin had an index between 0.4 and 0.49. Along with an advancement of ecological civilization construction, we should increase the Beijing–Tianjin–Hebei region's coordinated development and continuously strengthen its conservation and intensive land-resource use by increasing the forest area, forest coverage, the construction of national and provincial nature reserves, and ecological red lines. We should carry out space planning, promote the intensive utilization of its production spaces, make its living spaces pure, fresh and livable, and create ecological spaces with beautiful scenery. The overall benefits of green space have been improved, but in comparison, Tianjin, Hengshui and other cities had relatively low green space benefits due to high utilization rates of land development and large proportions of built-up areas.

## 4. Strategy for Improving Green Land Use Benefit Based on Evaluation

Through the evaluation of green land use benefits in the Beijing–Tianjin–Hebei region, the green land use benefits were not high, and the regional differences were obvious. The restricting factors of the green benefits were prominent. At present, it is imperative to focus on the strategic goal of green and high-quality development, to solve the prominent ecological and environmental problems of land use, to optimize the pattern of development and the utilization of territorial space resources, to promote coordinated regional development, to improve the green governance benefits, and to provide strong support for further promoting the coordinated development of the region.

### 4.1. Improve the Pattern of Development and Protection of Territorial Resources, and Systematically Improve the Green Land Use Benefit

Land use is a comprehensive system with a high degree of integration of natural resources, economy, society and the ecological environment. It is a comprehensive expression of the allocation of utilization modes, operation characteristics and coverage character-

istics of territorial space, and a concentrated reflection of the resource development and utilization pattern of a territorial space, such as land use control, farmland protection, ecological restoration and environmental governance. Under the concept of ecological priority and green development, we should attach great importance to the different regional characteristics of land use and its green benefits, to strengthen territorial space planning, to promote the optimization of the development and protection patterns of territorial space resources, and to improve the green benefits of land use.

(1) Adhere to the guide of green land use and improve green land use ecological benefits. Ecological priority green development is an important strategy for China's economic and social development, emphasizing green land use and deepening the implementation of green development. According to the above evaluation, the green land use ecological benefits in the Beijing–Tianjin–Hebei region contributes the most to the green land use benefits, but it still holds great development potential. The main factor affecting the level of green land use ecological benefits is the amount of ecological resource reserves. To improve the level of green land use ecological benefits, we should make full use of the advantages of the regional ecological resources, formulate a green and scientific land use plan, and protect the areas of high ecological value such as the wetlands, grasslands and forests. In areas with poor reserves of ecological resources and serious damage to the ecological environment, it is necessary to severely punish actions of over-exploitation, deforestation and reclamation, to establish land development and protection plans, to protect the regional ecological system, to improve the ability of self-regulation and repair of the ecological system, and to ensure the output of green land use ecological benefits.

(2) Promote the industrial ecology and ecological industrialization, and improve the green land use economic benefits. According to the above evaluation, the Beijing–Tianjin–Hebei region has significant room to improve its green land use economic benefits, while the advantages of its agricultural, forestry, animal husbandry and fishery resources have not been utilized. We should develop a leading green industrial system according to the local conditions, as well as promote the development of low-emission, circular and pollution-reducing industries, develop fishery industries in coastal areas, realize the large-scale development of animal husbandry in the plains areas and hilly areas, and form a forestry industrialization pattern through the construction of forestry bases. The development of agriculture and animal husbandry in urban areas is limited and urban modern agriculture should be promoted, so that the supply of agricultural products in urban areas can be guaranteed under the concept of green development; thus, improving the green economic benefits of the regional land areas.

(3) Strengthen guidance for territorial space planning and improve the green land use space benefits. In territorial space planning, green space is an important issue in the implementation of an ecological priority strategy. The Beijing–Tianjin–Hebei region has great potential for the development of green space benefits, and these can be fully brought into play by optimizing the territorial space. To improve the green land use space benefits, we should optimize the land green space patterns, improve the forest coverage rates and reduce the desertification areas. To ensure areas of nature reserves, we should formulate management plans for them and establish national and provincial nature reserves, strengthen the intensive use of land, strictly observe the ecological red lines, and promote the intensive and efficient use of production spaces. We should optimize the urban patterns, expand green park spaces, and rationally use the regional green spaces; thus, ensuring the green land use space benefits.

(4) Increase investment in a targeted manner and improve the green land use governance benefits. To realize green development, the state has been paying more attention to governance benefits and increasing its investment in urban environmental infrastructure in recent years. The green land use governance benefits in the Beijing–Tianjin–Hebei region contributes greatly to the green land use benefit, which is reflected in

the large investment of government funds and the significant effect of environmental governance measures. To improve the green land use benefit management, the government should pay attention to infrastructure construction and improve the centralized treatment rate of urban sewage. It should issue relevant policy documents and comprehensive controls of soil and water loss, and establish a comprehensive control system of those soil and water losses. It should focus on developing water-saving agriculture, optimizing agricultural water supply dispatching measures, and on ensuring effective irrigated land area. There is a need to improve the government's governance ability to ensure the green land use governance benefits.

### 4.2. Establish a Mechanism for Sharing Green Land Use Benefits and Promote Regional Coordination of Green Land Use Benefits

In China's action plan on establishing a market-based and diversified compensation mechanism for ecological protection, it is mentioned that the establishment of a mechanism for sharing green benefits is a key task in implementing green development. Under the influence of different resource endowments or different regional patterns, there are regional differences in their green benefits. The establishment of a green benefit sharing mechanism is conducive to the coordination of regional factors, promoting the complementarity of green benefits of regional land use and ultimately ensuring the continuous improvement of green benefits and the overall level of green land use benefits.

(1) Strengthen the development of regional green benefit communities and improve the capacity of green land use across the region. China has put forward the coordinated development strategy for the Beijing–Tianjin–Hebei region, built a community for the protection and utilization of ecological resources, adhered to the concept of green development and the principle of systemic integrity, and shared green benefits among regions. We should strengthen the cooperation between regional governments to break the administrative barriers of regional environmental protection and ecological product trade, to build trans-regional collaborative institutions for green benefit sharing, give full play to regional advantages and establish a regional economic green benefit community. A regional green circulation system should be built to promote the mutual benefits and complementarity of regional green benefits, to realize the regional coordination of green land use benefits and to improve the overall level of regional green land use.

(2) Promote the development of green land rights and the interest trading market and to promote land use green benefit trading. This involves establishing a green rights and interest trading market among the regions, taking their ecological resources as the common economic commodities and monetizing their green value via market trading. According to the green land use benefit evaluation index system, a green land use benefit accounting system was established, and the regional green benefit index was replaced to realize the regional sharing of green products of ecological resources and to ensure the overall improvement in the green benefits level.

(3) Strengthen the sharing of green benefits among stakeholders and promote the enthusiasm of green land use. Countries have put forward the construction of a "cost sharing, benefit sharing" ecological sharing mechanism of an ecological region and an ecological benefit compensation agreement. Poor regions can then share their ecological compensation costs and ecological service revenues, solving the problem of an unbalanced development area for green benefits, fully enhancing the enthusiasm for active ecological reserves, and promoting green land use in the region.

(4) Strengthen the supervision of green land use and establish a performance appraisal mechanism for green land use. In the monitoring and supervision of land use, according to the evaluation results of the green land use benefits, the performance assessment indicators of green land use benefits need to be defined, with unified supervision standards, targets and monitoring indicators being formulated, and green land use

development being strictly implemented to ensure the continuous improvement in green land use benefits.

## 5. Conclusions

Land use is a complex system of interaction between human activities and the natural ecosystem, comprehensively reflecting the coordinated development relationship between the economy, society, ecology, and the environment, and it reflects the degree of realization of the green development of land use modes, processes, and target outputs. Globally, green development is the inevitable choice for sustainable development. China has taken green development as its national strategy, and it is the only way for China to achieve high-quality development. It needs to implement the concept of green development into the process of land use, explore the impact of green land use changes suitable for the concept of green development on the soil organic matter, to promote a profound change in its land use patterns, and to provide a reference for the study of soil fertility, environmental protection and the global carbon cycle. This paper discusses establishing a green land use benefit evaluation index system and evaluation method, and it helps to focus consciousness on green land use and clear, green land use responsibility. This paper is helpful in promoting the green development of land use, providing theoretical guidance for the green performance assessment of that land use, and providing a decision-making reference for the design of a reward-and-punishment system. In this paper, the green development concept is input into the land use system, based on the integrity of the ecology, and social, economic and environmental land use properties, considering the natural, economic, ecological and social attributes of land use. Green benefits can be divided into green ecological benefits, green economy benefits, green governance benefits, and green space benefits. This paper takes the Beijing–Tianjin–Hebei region as the research area and constructs the evaluation index system of its land use green benefits. Combined with the entropy weight method and BP neural network method, the entropy-BP neural network model of green land use benefits was constructed to evaluate the green land use benefit in the region, and to reveal its regional pattern and influencing factors. The main conclusions are as follows: (1) The level of green land use benefit in Beijing–Tianjin–Hebei is still low. Among the four subsystems, the green space benefit index was the highest, followed by the green governance benefit and green economic benefit, while the green ecological benefit index was the lowest. (2) The degree of cultivated land, forest products, sewage concentrated treatment areas, respectively, had a significant influence on the green economic benefits, green ecological benefits, green governance benefits and green space benefits. The influence of arable land per capita, the per capita water resources, and the forest land area per capita, affected the level of the green ecological benefits. (3) The spatial pattern of the green benefits was high in the north and low in the middle. Langfang, Beijing and Handan were the low centers of the land's green ecological benefits, while Beijing and Tianjin were the low centers of the land's green economy benefits. The land's green governance and green space benefits were relatively balanced. Through the evaluation and analysis, it was revealed that the government should focus on the following two aspects to improve the green benefits of land use in the Beijing–Tianjin–Hebei region: (1) Adhere to the guide of green land use and improve the green land use ecological benefits; promote industrial ecology and ecological industrialization and improve the green land use economic benefits; strengthen the guidance for territorial space planning, improve the green land use space benefits, increase its investment in a targeted manner; improve the green land use governance benefits. (2) Strengthen the development of regional green benefit communities and improve the capacity for green land use across the region; promote the development of the land's green rights and an interest trading market; promote land use green benefits trading; strengthen the sharing of green benefits among the stakeholders; promote an enthusiasm for green land use; strengthen the supervision of green land use; establish a performance appraisal mechanism for green land use.

**Author Contributions:** Project administration, W.P.; writing—original draft preparation, Y.S.; data curation, Y.L.; writing—review and editing, X.Y. All authors have read and agreed to the published version of the manuscript.

**Funding:** This research was funded by the Beijing Social Science Foundation Major project "Research on building an ecological civilization system (19ZDA03).

**Institutional Review Board Statement:** Not applicable.

**Informed Consent Statement:** Not applicable.

**Data Availability Statement:** All data and models generated or used in the research process of this paper are presented and explained in the body of the article.

**Conflicts of Interest:** The authors declare no conflict of interest. There are no circumstances in which personal circumstances or interests could be deemed to unduly influence the reported findings. There are no funders for this article.

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
