# Peer review of "An Evaluation and Promotion Strategy of Green Land Use Benefits in China: A Case Study of the Beijing–Tianjin–Hebei Region"

_land, doi:10.3390/land11081158_

Round 1

Reviewer 1 Report

This paper is fairly well written and referenced. However, the following suggestions and corrections may improve the quality of this manuscript:

1.         Two keyword duplicates the same as in the paper title. Other selections should be re-chosen.

2.         The paper title should add the country name.

3.         All acronym names, such as BP, should be defined when first appear and be used thereafter.

4.         The abstract as well as the conclusion should emphasize more on the practical implications and contributions of this study.

5.         The word “green” has been used too many times to describe all the terminologies in this paper. What exactly is green technology? Is it the most appropriate technology to use?

6.         The first paragraph in the methodology section is more appropriate to be included in the introduction section.

7.         The captions for Figure 1 and Table 1 are not easy to understand.

8.         What is “dispersion degree” in Line 174?

9.         What is the meaning of the “Index type” in the fifth column of Table 1?

10.     Please explain “yuan” to the international readers.

11.     Nei Mongolia should be Inner Mongolia instead.

12.     How to standardize data to between 0 and 1?

13.     Unify “%” and “percent” or “per cent”.

14.     Again, explain “mu” to international readers. In fact, use metric unit throughout the entire text, such as ha.

15.     Clarity of Figure 2 is rather poor.

16.     Section 4 on “Promotion Strategy” seems to be recommendations from the authors. Have to support these strategies with scientific evidence, facts or references to convince readers they will improve green land use in the study areas.

17.     Speculation that this study results will work well in other locations may demonstrate significant contribution from this study.

18.     If possible, a geographic map of the study areas in respect to China may be helpful to foreigners.

19.     Many spacing, capitalization (pearl river), an d most importantly English language errors have been detected.

Author Response

Dear Editor,

    Thank you very much for giving us the opportunity to re-submit the revised manuscript Evaluation and promotion strategy of green land use benefit in China: A case study of Beijing-Tianjin-Hebei Region, may be published in Land as a research paper. The comments on our original manuscript were very insightful and constructive. We greatly appreciate the time and effort you and the reviewers have given to provide valuable feedback on our work. We revised and improved the manuscript according to the comments and gave detailed replies to each reviewer's comments. Attached is our revised manuscript, named "Manuscript-reviewer1". The first part is the revised manuscript, and the second part is our point-by-point responses to all comments.

    We hope that the careful revision of the manuscript and our accompanying response will meet land's academic publishing requirements. If you have any questions, please do not hesitate to contact me. We look forward to your early reply.

Best regards,

Yue Sun

Reviewer 2 Report

This paper used entropy weight method and BP neural network model to establish the evaluation model of green land use benefit in Beijing-Tianjin-Hebei region, and explore the optimization path of land use in Beijing-Tianjin-Hebei region and the policy measures for the coordinated development of land use and ecological environment. It can provide reference for the green and high-quality development of land management in the region to some extent, but the following points still need to be improved.

1. The summary section, "The change of land use and land cover will comprehensively change the land benefit, which reflects the green land use mode and green land use benefit." is puzzling and abrupt. Are land use and land cover changes so closely linked to land benefits? Or does it simply reflect green land use model and green land use benefits?

2. The second paragraph of the introduction lists previous studies that have analyzed the benefits of green land use from various perspectives but are not well articulated. What are the links between these studies and how has research in this area progressed? (content, methods, data...)

3. The third paragraph of the introduction talks about China's important contribution to global ecological security and sustainable development, but it is not specified below, and some of the articles cited are not from China, how does it argue that China makes an important contribution?

4. The third paragraph of the introduction piles up the different methods used by many existing studies to assess land use change and its associated benefits methods, but does not point out the advantages or shortcomings of the methods used in this paper (entropy weight method and BP neural network models) compared to these methods.

5. What is the output value/target value of the BP neural network model? The article is silent on this.

6. The methodology section is large and redundant, and much of it is conceptually defined here and should be briefly described as background in the introduction.

7. The study area section only introduces the physical geographic characteristics, socio-economic, and resource environment of Beijing-Tianjin-Hebei, but does not explain why this region is chosen as the study area, and does not clarify the necessity of studying the benefit of green land use in this region.

8. Can the evaluation system constructed in the article effectively assess the benefit of green land use? Is there any theoretical source or literature to support it.

9. Figure 1 and Table 1 are missing names.

In addition, there are several related papers that may be helpful for your introduction section:Cities are going uphill: Slope gradient analysis of urban expansion and its driving factors in China. Science of the Total Environment; Multi-scenario simulation of urban land change in Shanghai by random forest and CA-Markov model; Spatio-temporal evolution and influencing factors of urban green development efficiency in China. Journal of Geographical Sciences.

Author Response

Dear Editor,

    Thank you very much for giving us the opportunity to re-submit the revised manuscript Evaluation and promotion strategy of green land use benefit in China: A case study of Beijing-Tianjin-Hebei Region, may be published in Land as a research paper. The comments on our original manuscript were very insightful and constructive. We greatly appreciate the time and effort you and the reviewers have given to provide valuable feedback on our work. We revised and improved the manuscript according to the comments and gave detailed replies to each reviewer's comments. Attached is our revised manuscript, named "Manuscript-reviewer2". The first part is the revised manuscript, and the second part is our point-by-point responses to all comments.

    We hope that the careful revision of the manuscript and our accompanying response will meet land's academic publishing requirements. If you have any questions, please do not hesitate to contact me. We look forward to your early reply.

Best regards,

Yue Sun

Round 2

Reviewer 1 Report

A good job in revising the original manuscript. Most of my previous comments and suggestions have been adequately addressed. The revised version is much improved. 

With a thorough proof check and English language check. The revised manuscript is readied for publication acceptance.

Author Response

Dear Editor,

Thank you very much for giving us the opportunity to re-submit the revised manuscript (land-1810590) “Evaluation and promotion strategy of green land use benefit in China: A case study of the Beijing–Tianjin–Hebei Region”, may be published in Land as a research paper. Thank you very much for your suggestion of English revisions. We use English editing services at links https://www.mdpi.com/authors/english for syntax and presentation corrections. The edited words/sentences in red in the resubmitted manuscript file. The resubmitted manuscript file name is “Evaluation and promotion strategy of green land use benefit in China- A case study of the Beijing–Tianjin–Hebei Region(manuscript-round2)”. 

We hope that the careful revision of the manuscript and our accompanying response will meet Land's academic publishing requirements. If you have any questions, please do not hesitate to contact me. We look forward to your early reply.

Best regards,

Wenying Peng

E-mail:  cjpengwy@cueb.edu.cn
